



# Firn Seismic Anisotropy in the North East Greenland Ice Stream from Ambient Noise Surface Waves

Emma Pearce[1], Dimitri Zigone[1], Coen Hofstede[2], Andreas Fichtner[3], Joachim Rimpot[1], Sune Olander Rasmussen[4], Johannes Freitag[2], and Olaf Eisen[1,2,5]

[1]University of Strasbourg/CNRS, Institut Terre and Environment of Strasbourg (ITES), UMR7063, Strasbourg, France
[2]Alfred Wegener Institute, Helmholtz Centre for Polar and Marine Research, Bremerhaven, Germany
[3]ETH Zürich, Zürich, Switzerland
[4]Centre for Ice and Climate, Section for the Physics of Ice, Climate and Earth, Niels Bohr Institute, Copenhagen, Denmark
[5]University of Bremen, Bremen, Germany

**Correspondence:** Emma Pearce (epearce@unistra.fr), Olaf Eisen (Olaf.Eisen@awi.de)

**Abstract.** We analyse ambient noise seismic data from 23 three-component seismic nodes to study firn velocity structure and seismic anisotropy near the EastGRIP camp along the Northeast Greenland Ice Stream (NEGIS). Using 9-component correlation tensors, we derive dispersion curves of Rayleigh and Love wave group velocities from 3 Hz to 40 Hz. These velocity distributions exhibit anisotropy along and across the flow. To assess these variations, we invert dispersion curves for shear wave velocities ($V_{sh}$ and $V_{sv}$) in the top 150 m of NEGIS using a Markov Chain Monte Carlo approach. The reconstructed 1-D shear velocity model reveals radial anisotropy in the firn, with $V_{sh}$ 12%–15% greater than $V_{sv}$, peaking at the critical density (550 kg m$^{-3}$). We combine density data from firn cores drilled in 2016 and 2018 to create a new density parameterisation for NEGIS, serving as a reference for our results. We link seismic anisotropy in the NEGIS to effective and intrinsic causes. Seasonal densification, wind crusts, and melt layers induce effective anisotropy, leading to faster $V_{sh}$ waves. Changes in firn recrystalisation cause intrinsic anisotropy, altering the $V_{sv}$ to $V_{sh}$ ratio. We observe a shallower firn-ice transition across flow ($\approx 50$ m) compared to along flow ($\approx 60$ m), suggesting increased firn compaction due to the predominant wind direction and increased deformation towards the shear margin. We demonstrate that short-duration (nine-day minimum), passive, seismic deployments, and noise-based analysis can determine seismic anisotropy in firn, and reveal 2-D firn structure and variability.

## 1 Introduction

Firn forms when snow survives for more than one year. When the yearly new snow accumulation increases the overburden pressure, the snow properties transition into firn and finally, pure glacier ice (Herron and Langway, 1980). The compaction profile of the firn column therefore preserves the local climate history, recording the amount of snow accumulation, snow melt, temperature conditions and the subsequent preservation of the snow. Firn also represents an important factor of uncertainty for estimating changes of surface mass balance (Verjans et al., 2021; Kowalewski et al., 2021), where the current rate of change of ice sheets' mass balance is often derived from tracking changes in surface elevation from satellite altimetry or interferometry. For a correct interpretation, potential changes in firn densification rates over time and space have to be accounted for. Firn



is also used for monitoring ice sheet collapse (Hubbard et al., 2016), preservation and close-off depth of atmospheric gas in bubbles relevant for paleoclimate reconstruction and ice core interpretation (Újvári et al., 2022; Schwander et al., 1997) and correcting for near-surface effects in radar (Mojtabavi et al., 2022), and seismic data (Smith et al., 2021).

The variations of the mechanical properties of firn in principle depend on ice crystal anisotropy, which influences ice flow on larger scales (Duval et al., 1983; Bons et al., 2016). Ice crystal preferred orientation axes are shown to already be affected within the shallow subsurface. At EastGRIP camp on the North East Greenland Ice Stream (NEGIS), measurements of ice crystal eigenvectors show a clear characterisation of crystal anisotropy already by a depth of 100 m (Weikusat; Gerber et al., 2023). Although not showing anisotropy in the firn, it strongly indicates that the fabric is being generated early. Recently, Oraschewski

and Grinsted (2022) and Grinsted et al. (2022) showed for the shear margin of the NEGIS that the firn densification is a function of the applied strain and higher in the shear margin. Hence there is likely firn anisotropy present in the NEGIS.

Most direct knowledge about the vertical distribution of anisotropy in firn and ice has been obtained from a few cores in Greenland and Antarctica (e.g. Vallelonga et al., 2014). However, these analyses are very laborious, and only provide discontinuous 1-D information along the vertical axis, and usually have very coarse resolution. In addition, given the fragility

of firn cores, anisotropic properties of firn, for instance, obtained by standard thin-section fabric analyses, are rare. In 2023, a new standardised densification model of EastGRIP has been created using core data from 2016 and 2018. These data are presented here, and used for comparison with our ambient seismic noise dispersion analysis results.

Seismic waves offer a means to image firn and ice structures that complement the capabilities of other techniques. For example, in glaciology, seismic methods have been shown to be an effective tool to determine firn structure and velocity

profiles, from the use of refraction data (e.g. Schlegel et al., 2019; Hollmann et al., 2021; Pearce et al., 2023; Picotti et al., 2015), distributed acoustic sensing methods (Zhou et al., 2022; Fichtner et al., 2023), or from ambient noise, (e.g. Chaput et al., 2022; Lévêque et al., 2010; Diez et al., 2016; Zhang et al., 2022). The use of seismic methods can give a unique insight into the firn anisotropy that is otherwise difficult to obtain.

Major progress has been made over the last decade to understand and partly identify anisotropic areas from the ice surface by

geophysical methods. Among these methods, noise correlation techniques have been used. Diez et al. (2016) retrieved Rayleigh and Love waves from noise and inverted them for shear velocity ($V_{sh}$ and $V_{sv}$) profiles at the Ross Ice Shelf. Zhang et al. (2022) uses seven days of ambient-noise data to obtain 2-D vertical and horizontal shear-wave velocity models. Azimuthal anisotropy within firn has been explored using DAS by Zhou et al. (2022). However, their results show large uncertainty, and hence, they could not rule out anisotropy in the firn layer. Instead, they suggest the use of ambient noise surface waves tomography to

investigate this further.

Ambient noise tomography is a widely adopted technique in seismology for obtaining high-resolution tomographic images spanning kilometre-scale dimensions (see Campillo and Roux (2015) and Shapiro (2019) for reviews). Unlike conventional methods reliant on impulsive sources such as earthquakes or active seismic shots, noise-based imaging utilises the diffuse and random nature of the wave-field to reconstruct virtual active sources at every passive station by cross-correlating the continuous

seismic noise records between every pair of stations (Shapiro et al., 2005).



Thanks to its simplicity this technique has revolutionised the use of seismic arrays for imaging complex structures at various scales such as fault zones (e.g. Zigone et al., 2015, 2019), sedimentary basins (e.g. Schippkus et al., 2018), urban areas (e.g. Lin et al., 2013), geothermal reservoirs (e.g. Lehujeur et al., 2018), landslides (e.g. Renalier et al., 2010), water catchments (e.g. Wang et al., 2019), mountain glaciers (e.g. Sergeant et al., 2020) and ice shelves (e.g. Diez et al., 2016; Zhang et al., 2022).

With the advancement of seismic nodes (a compact all-in-one 3-channel, wireless seismometer), small dense seismic arrays are being deployed in polar regions (e.g. Gimbert et al., 2021; Chaput et al., 2022; Zhang et al., 2022), giving rise to the possibility of using ambient noise-based tomography to study shallower structures and anisotropy. The most notable advantages of this method are that ambient seismic noise exists in all places, (with varying spatial and temporal variations), and that the method utilises a passive acquisition with relatively easy deployment.

Here we present passive seismic data acquired over 29 days on the NEGIS. The corresponding noise-based Love and Rayleigh wave dispersion curves are inverted to estimate the S-wave velocity structure of the top 150 m of the NEGIS and the radial anisotropy within the firn (variations between horizontally and vertically polarised shear wave velocities). Our findings contribute novel insights into the internal structure of the NEGIS and advance our understanding of the complex dynamics characterising firn and ice formation and show that environmental noise seismic acquisition is a useful and simple tool to
obtain local firn properties.

## 2   Study Area

The NEGIS is Greenland's largest ice stream. To investigate its internal properties, the International Partnership for Ice Core Sciences (Dahl-Jensen et al., 2021) proposed to drill an ice core in the onset region. Since 2015 an ice core has been drilled by the EastGRIP consortium. The drilling reached bedrock in July 2023, at approximately 2690 m depth. In the vicinity of the
EastGRIP drill site (N 75.63, W -35.98 in 2022), 500 km from the calving front of the ice stream, we deployed a network of 23 three-component (N, E, and Z) SmartSolo seismic nodes (Fig 1).

The ice flow direction at EastGRIP is approximately 20° from North (NNE) with a speed of approximately 60 m per year (Joughin et al., 2018; Grinsted et al., 2022). Nodes were deployed along flow (Fig. 1c, Line 1, 20° from North), across flow (Fig. 1c, Line 5, 110°from North) and between (42.5°, 65° and 82.5° from North), at varying distances from 100 m to 4200 m
away from the drill site.

The crystal fabric in the ice of the NEGIS is well known in this area, with direct measurements along the ice core from 100 m depth down showing a broad vertical single maximum (largest eigenvalue in the vertical, $\lambda_1 = 0.25$, $\lambda_2 = 0.25$, $\lambda_3 = 0.5$). From 100-200 m, a transition from this broad single maximum into a broad vertical girdle occurs by redirecting the c-axis from the vertical into the horizontal across-flow plane (Gerber et al., 2023), which then becomes the largest eigenvalue. Gerber et al.
(2023) joined a suite of geophysical observations and methods with numerical modelling to map the spatial distribution of the fabric across the NEGIS.



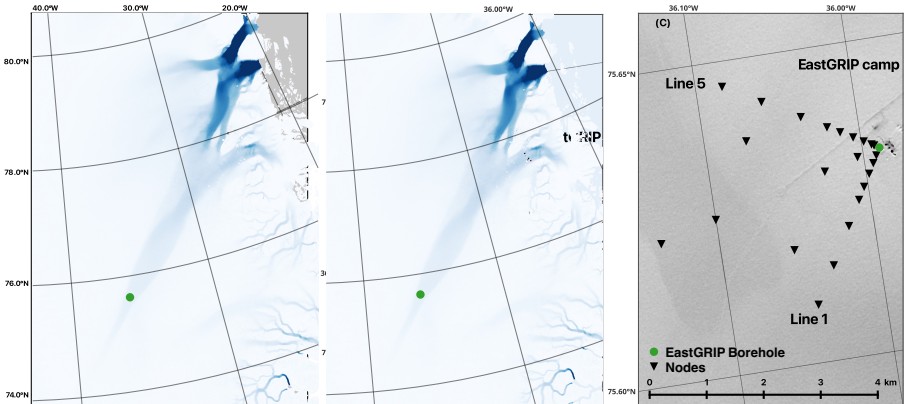

**Figure 1.** Location of seismic node array. (A) Surface ice flow velocity of the Greenland Ice Sheet from Sentinel-1 (2019–2020) (Joughin et al., 2018), highlighting the Northeast Greenland Ice Stream (NEGIS) and its three major outlet glaciers. (B) The seismic node array that was used in 2022 to acquire 29 days of ambient noise data beginning 21 July 2022 and the location of the two cores obtained in 2016 and 2018 used for the standardised density parameterisation at EastGRIP. (C) Close-up of the camp area showing the location of the drilling and other camp infrastructure (trench, storage, airfield, etc.) and the location of the seismic nodes. Satellite image from LandSat 8.

In addition to the main EastGRIP core density data obtained in 2016, in 2018, core S6 was acquired close to the borehole (1.2 km away). We combined the available density data, and include the first interpretation of a standardised EastGRIP density parameterisation, which we will use as a reference for our results inferred from passive seismic.

## 3 Data, Noise Processing and Cross-Correlations

We use continuous noise data from 23 stations recorded during 2022 from July 21 until August 21 with IGU-16HR 3C 5Hz SmartSolo nodes. The network recorded at a sampling rate of 1000 Hz, with inter-station distances ranging from $\approx$ 50 m to 2900 m. We process the ambient noise data in the following order; (1) single station data preparation, (2) cross-correlation and temporal stacking, (3) measurement of dispersion curves and (4) quality control, including error analysis and selection of group velocity measurements. Processing of ambient noise data follows a typical processing chain (Bensen et al., 2007; Poli et al., 2013) but modified for higher frequency content (see Zigone et al., 2019).

### 3.1 Data pre-processing

Pre-processing steps are used on the raw data to increase the quality of the correlation functions and subsequently, dispersion measurements.

The following signal processing is done for each individual station.

1. The data are down-sampled to 100 Hz and high-pass filtered at 0.04 Hz.



2. The data are split into 15-minute segments: for each segment, a first glitch correction, which consists of a clipping at 15 standard deviations is performed to remove any high amplitude samples caused by the digitisation. Additionally, we remove segments where gaps exceed 10% of the segment length. Finally, segments with an energy greater than 3.5 times the mean energy over the entire day are removed, as this suggests the presence of a transient source signal (e.g. ice quake) within the segment.

3. The remaining segments are then whitened by dividing the amplitude of the noise spectrum by its absolute value for frequencies between 0.1 Hz and 50 Hz, the Nyquist frequency (Bensen et al., 2007).

4. Finally, a second clipping of the data at 3 standard deviations is performed to further dampen the remaining transient signals that may have been missed with the energy threshold.

## 3.2 Correlations

With the processed 15-minute segments, we compute the cross-correlation function between all station pairs in the frequency domain, as by Bensen et al. (2007). All correlation functions from all segments in a day are then averaged to obtain a single daily correlation function per station pair. As all three components (vertical (Z), North-South (N) and East-West (E)) are recorded by the stations, we compute the nine inter-component correlation functions corresponding to the elastic Green's tensor (ZZ, ZE, ZN, EZ, EE, EN, NZ, NE, NN).

Figure 2 shows an example of the ZZ correlation functions between two stations with 850 m inter-station distance, presented as a correlogram. We first note a high-frequency arrival at zero correlation time. We associate this feature with high-frequency wind resonance in the two-foot bamboo sticks that sat 20 cm above the snow, used to mark the location of the seismometers. Such near-zero arrivals are sometimes observed on ZZ correlation functions, especially at high frequencies, and do not prevent the use of the later arriving surface waves for imaging purposes (e.g. Zigone et al., 2019). In Fig. 2 the surface wave is clearly observed in positive time around 0.5 s. Even if some small variations are visible in the daily correlation functions – for example the decrease of amplitudes between Julian days 211 and 214 – the surface wave arrival remains stable for most of the 29 days recording period.

We note that our correlation functions are asymmetric with much higher amplitudes in positive times. If sources of ambient noise are distributed homogeneously in azimuth, the causal (the positive lag) and acausal (the negative lag) signals would be identical. However, asymmetry in amplitude and spectral content is typically observed, as in our data, which indicates differences in both the source process and distance to the source in the directions radially away from the stations. The asymmetry in our correlation functions likely comes from the fact that most of the human activities at EastGRIP are concentrated around the main camp and the borehole trench (see Fig. 1c) leading to more energy in the positive times, for propagation away from the borehole trench. Such asymmetry in Green's functions retrieval is typical of noise-correlation studies as homogeneous noise sources are rarely possible, except in specially designed experiments with controlled noise-source distributions (e.g. Roux et al., 2004). Our study faces a similar challenge, where the noise-source distribution is non-uniform, which may bias the measured travel times on correlation functions (e.g. Froment et al., 2011). However Yao and Van Der Hilst (2009) and Hillers et al. (2013)





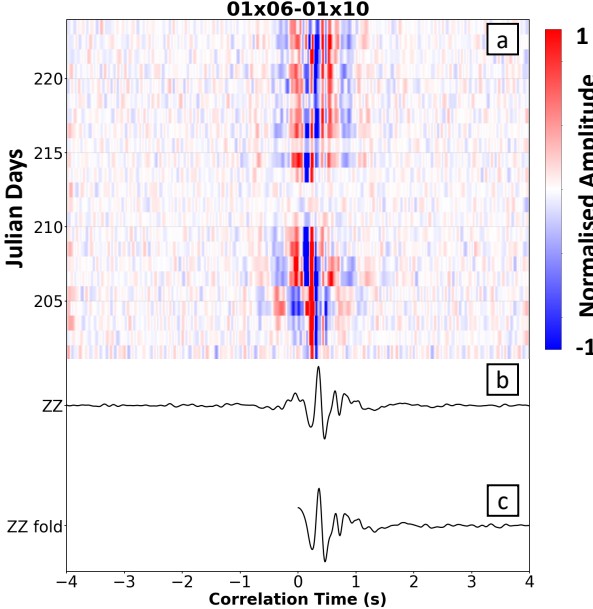

**Figure 2.** (a) Daily ZZ correlation functions plotted as a correlogram for station pair 01x06-01x10 (an inter-station distance of 850 m). (b) Stack of the ZZ correlation functions for the whole month, and (c) the final folded ZZ correlation function for station pair 01x06-01x10. Data are filtered between 1 Hz and 7 Hz.

studied the potential errors in arrival-time measurements of Rayleigh waves due to the directional noise and found the effect
to be small. In addition, to better sample the sightly varying noise sources distribution around EastGRIP during the month of
recordings, we stacked the 29 daily correlation functions. The stack also aims to reduce temporal variations in the correlations
that can affect the quality of the cross-correlations, thus improving the SNR. Finally, to broaden the frequency content, we fold
the cross-correlations to merge low and high-frequency information for the following travel time measurements (e.g. Shapiro
and Campillo, 2004) (see Fig. 2).

After stacking, the elastic Green's correlation tensor is rotated along the inter-stations azimuth to provide the correlation
functions between the radial (R), transverse (T), and vertical (Z) components (RR, RT, RZ, TR, TT, TZ, ZR, ZT, ZZ) of
the seismic wave field propagating between the stations. This is illustrated in Fig. 3 which presents the nine components of
the correlation tensor as a function of inter-station distances. The arrival patterns observed are dominated by surface waves
travelling between the pairs of stations used. This is indicated by the visible frequency dependence of the reconstructed waves
velocity which is the dispersive characteristic of surface waves. Prominent Rayleigh waves are observed on the RR, ZZ, RZ,
and ZR components and Love waves are visible on the TT correlation term. Note that some scattered energy is still visible on
mixed transverse terms (ZT, TZ, RT and TR) due to complex interactions between the wave-field and firn layers as observed
in other complex structures such as fault zones (e.g. Zigone et al., 2019).





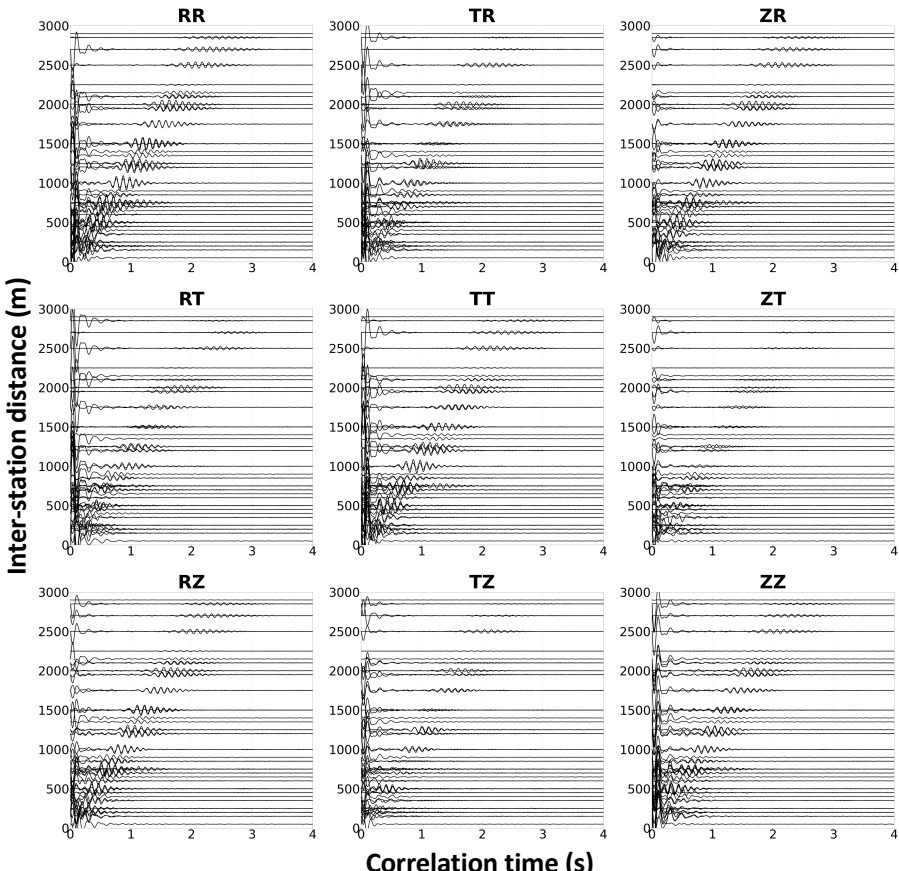

**Figure 3.** Nine inter-component correlation tensor for all station pairs filtered between 3 and 10 Hz. ZZ, RR, ZR, RZ components are dominated by dispersive Rayleigh waves, while the TT component shows dispersive Love waves.

## 3.3 Dispersion Curves

We measure group velocity dispersion curves for all station pairs and then invert them to obtain shear wave velocity 1-D depth profile. Dispersion measurements are performed on the folded correlation functions (see Fig. 3) for frequencies between 1 Hz and 50 Hz using the automated multiple filter technique of Pedersen et al. (2003).

For Rayleigh waves, we utilise the four components of the correlation tensor (RR, ZZ, RZ, ZR) that contain Rayleigh waves, and for Love we use only the TT component. We first run the automated multiple filter technique for each component, independently, to obtain a normalised frequency–group velocity diagram. The results for RR, ZZ, RZ and ZR are then combined with a logarithmic stacking in the frequency–group velocity domain as in Zigone et al. (2015). We then stack the dispersion diagrams (independently for Love and Rayleigh waves) for station pairs along flow and across flow producing four dispersion diagrams on which we took the maximum of the amplitude at each frequency as being the dispersion curve (Fig. 4 and 5). To



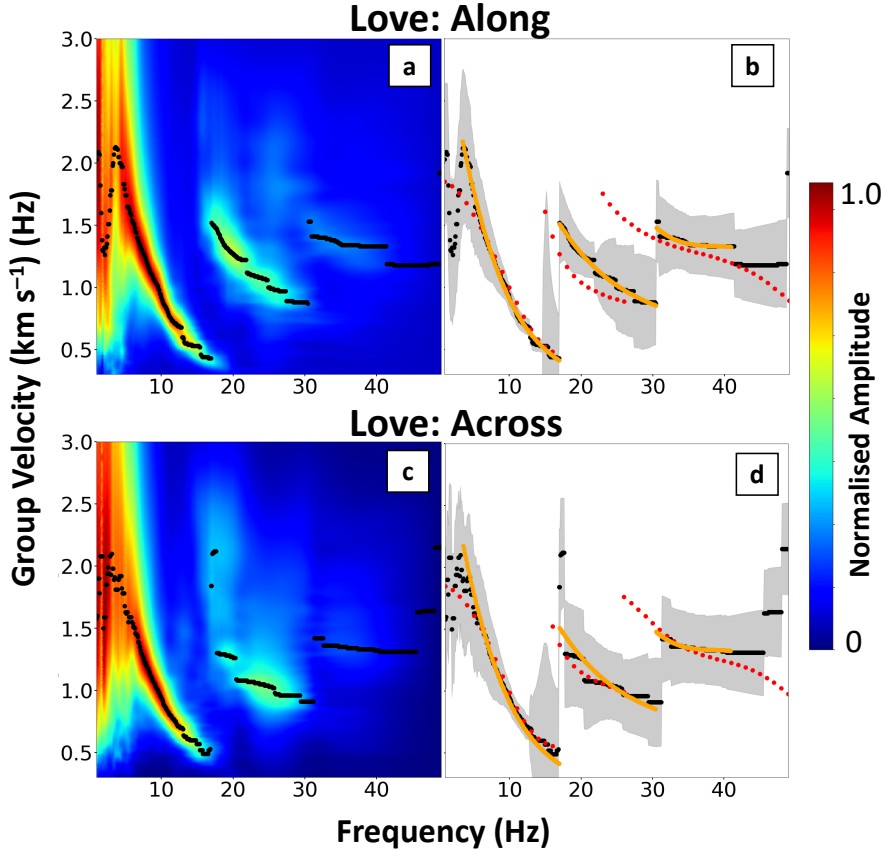

**Figure 4.** Left: stacked dispersion diagram for Love waves for (A) along and (C) across flow stations, with the maximum amplitude shown by the black dots. Right: The maximum group velocity (black) and error (± one standard deviation). Orange shows the smoothed values used as the input for Markov Chain Monte Carlo inversion. Red dots are the forward-modeled dispersion curves.

avoid anomalous measurements, we further remove the frequencies for which the dispersion measurements look anomalous. The remaining values selected for the inversion are presented as orange curves on the right of Fig. 4 and 5.

Our dispersion measurements all show the fundamental mode, with some also showing the first and second harmonic modes that introduce higher frequency data. For our final inversions (see details in section 4), we use group velocity values from all modes that can be properly identified. To confirm we are selecting group velocities for the correct mode, we first run the Vs

inversion with only the fundamental mode. The resulting velocity model is then used to compute synthetic dispersion curves to inform the selection of the first and when possible second higher modes using the same selection criteria as for the fundamental mode (see Fig. 4 and 5).

To obtain the error estimates for our group velocity values, we fit the dispersion diagram at each selected frequency with a Gaussian curve and assume the velocity error to be ± one standard deviation. Figures 4b,d and 5b,d show the selected

dispersion curves used for the Love and Rayleigh waves respectively.





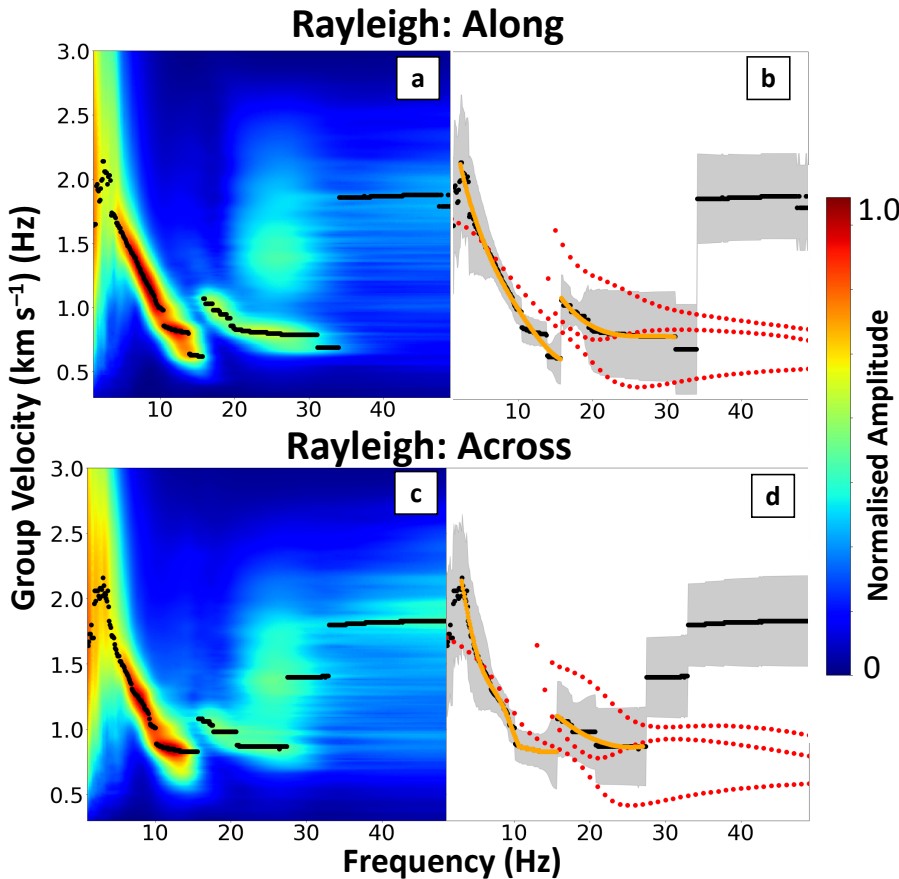

**Figure 5.** Left: stacked dispersion diagrams for Rayleigh waves for (a) along and (c) across flow stations, with the maximum amplitude shown by the black dots. Right: The maximum group velocity (black) and error ($\pm$ one standard deviation). Orange shows the smoothed values used as the input for MCMC inversion. Red dots are the forward-modeled dispersion curves obtained from a velocity model generated using only the fundamental mode.

## 3.4 Density data and parameterisation

A density parameterisation of EastGRIP has been created and is used for the first time in our discussion. The data used to create the parameterisation were obtained from various locations close to the EastGRIP main core. Firstly, from the EastGRIP main borehole in 2016 to a depth of 117.15 m. Secondly, two, 5 m deep trenches at the EastGRIP camp, and finally, a 72 m long

shallow core (denoted S6) drilled in 2018. For the top 5 m, liners (thin-walled carbon fibre tubes) were used to retrieve 100 one-meter long firn sections with less mechanical influence than what is possible during drilling. The values were averaged to produce one density data point for each one-meter interval from the surface to five meters depth. From the S6 shallow core, the firn-core segments have a lot of scratches in the section above 12 m, leading to likely underestimation of the density, and therefore, we only use data from 12 meters and downwards in one-meter resolution. The EastGRIP main core density record



has a 0.55 m resolution and covers the interval 13.75 — 117.15 m. All data points where the measurement notes indicate missing parts or bad data quality (producing outliers only in the downward direction) have been excluded. At 117.15 m depth, the density is about 900 kg m$^{-3}$. In order to provide a standardised density profile and allow the calculation of extrapolated density for deeper sections, a density parameterisation was derived based on all data points. The parameterisation also allows the calculation of the ice-equivalent depth at all depths. The details of the parameterisation are provided along with the density

data sets at PANGEAE.

    A visual inspection of three-dimensional reconstructions of pore and ice structure along the firn column from core S6, measured by X-ray tomography (Freitag et al., 2013), reveals no significant geometrical anisotropy in pore and ice structures. Enclosed air bubbles below the firn ice transition at around 65 m depth show no indications for systematic deviations from spherical shapes (Fig. 6). Layering from a few centimetre thick layers of different densities in the core is strong down to about

50 m followed by a decreasing trend downwards, with the addition of 1 mm sized wind crust layers occurring one to two times per meter.




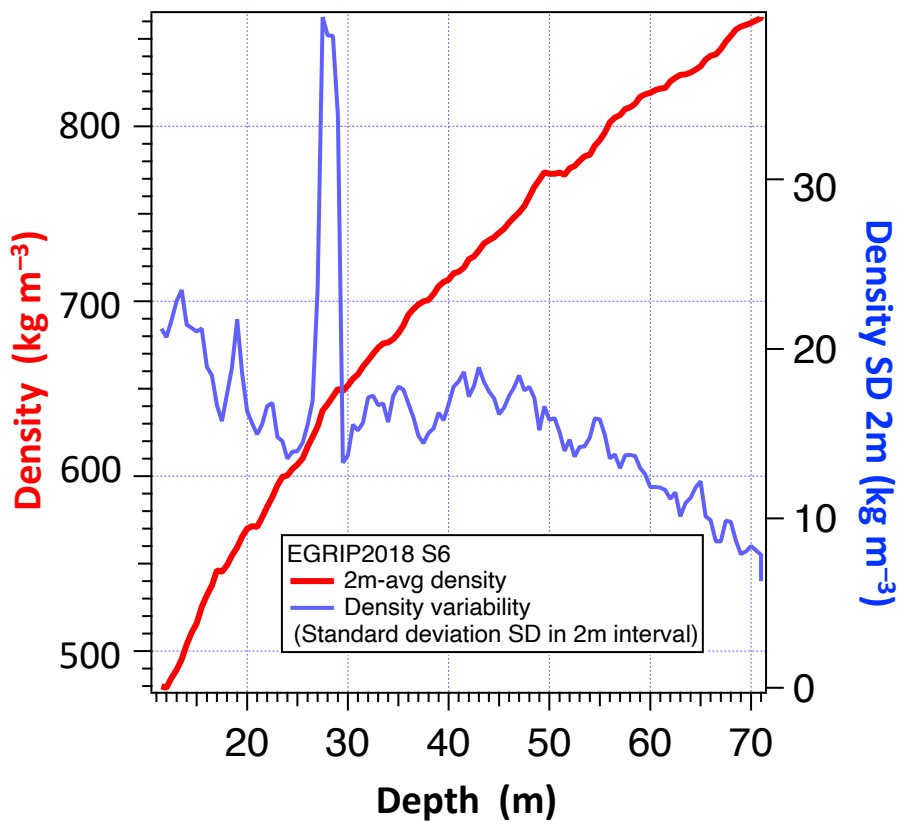

**Figure 6.** Plotted in blue, the evolving standard deviation (SD) of the density in 2 m intervals along the depth profile, averaged density in red. The density profile is measured by X-ray absorption in 0.1 mm vertical resolution. The spike in the SD profile is due to a prominent refrozen melt layer in 28 m depth with a strong density jump of more than 250 kg m$^{-3}$.

## 4 Results: Inversion of Dispersion Measurements for Shear Wave Velocity

We use a Metropolis Markov Chain Monte Carlo (MCMC) inversion (Brooks, 1998) to invert our dispersion measurements for $V_{sv}$ $V_{sh}$. The MCMC approach is used to solve nonlinear inverse problems with non-unique solutions (e.g. Sambridge and

Mosegaard, 2002; Shapiro and Ritzwoller, 2002; Zigone et al., 2019; Gallagher et al., 2009; Lehujeur et al., 2018).

We use a parameterisation with six layers over a half-space and invert for the S-wave velocity and depth of each interface (parameters given in Table 1). The computation of the logarithm of the likelihood for each model (i.e. the misfit function) is done with a linear combination of the logarithm of the prior density functions from both the model and data spaces (see Lehujeur et al. (2018) and Zigone et al. (2019) for more details). In the context of this study, the prior probability density

function (PDF) of the model space is determined by taking the product of uniform PDFs for each parameter present in the model. The prior PDF of the data space is approximated by employing lognormal probability distributions centred around the velocity values obtained from the dispersion curve. We consider that each dispersion point is independent which means



**Table 1.** Parametrisation used for the MCMC inversion.

| Layer number | Top depth range (m) | Vs range (km/s) |
|---|---|---|
| 1 | 15 | 0.1–1.9 |
| 2 | 30 | 0.1–1.9 |
| 3 | 45 | 0.1–1.9 |
| 4 | 80 | 0.1–1.9 |
| 5 | 100 | 0.1–1.9 |
| 6 | 150 | 0.1–1.9 |

that the covariance matrix utilised for the data space is taken to be diagonal (Tarantola, 2005). The forward modelling of the dispersion curves for each possible model is done with the program by Herrmann (2013) which use a modal summation method

(Herrmann, 2013). For each stacked dispersion curve, we run 15 independent Markov chains in parallel in which the step from one model to the next is governed by a Gaussian proposal PDF with a diagonal covariance matrix. Frequency ranges, and modes used for each model are given in Table A1. The terms of that covariance matrix are adjusted along the inversion to stabilise the acceptance ratio around 25%. Each chain runs until 1,500 models have been accepted. In total, the inversion keeps about 22,500 models over the 90,000 were tested for each depth profile. From those 22,500 models, the median of the 1,000

best models is taken as the solution of the inversion. Using the selected group velocity values for Love and Rayleigh, with an error of $\pm$ 1 standard deviation, we run separate inversions for each stacked dispersion curve, resulting in six independent shear wave velocity models for the NEGIS (Fig. 7) These models provide an averaged 2D velocity for the extent of the along and across flow lines.

In our results (Fig. 7) we observe two key features. Firstly, that $V_{sh}$ is always greater than $V_{sv}$. Secondly, that the magnitude

of the difference between $V_{sh}$ and $V_{sv}$ changes with depth.

The variation of $V_{sh}$ with direction (Fig. 7a) show indistinguishable changes between the along and across flow directions above 60 m. Below this, the across flow direction has higher velocities. In Fig. 7b, $V_{sv}$ shows more variation in velocity with direction and has larger uncertainty due to the dispersion curves for Rayleigh waves having a lower maximum frequency (Table A1).

Both along and across flow consistently have $V_{sh}$ greater than $V_{sv}$ (Fig. 7d,e) resulting in a positive radial seismic anisotropy $\left(\frac{vsh}{vsv} - 1 \times 100\right)$ at all depths, with the along flow direction having $V_{sh}$ and $V_{sv}$ velocities closer together than the across. The value of the radial anisotropy varies with depth, reaching a maximum of 15% for across flow, and 12% for along flow at a depth of 30 m, with the ratio then decreasing with depth (Fig. 7c).

Because the frequencies used in our inversions are limited to around 30 to 40 Hz depending on inversions, we have limited

sensitivity to the top 10 m – 20 m of the model (see Rayleigh and Love waves sensitivity kernels on Fig. AA1). Hence, we can make no conclusions about the radial anisotropy in this part.





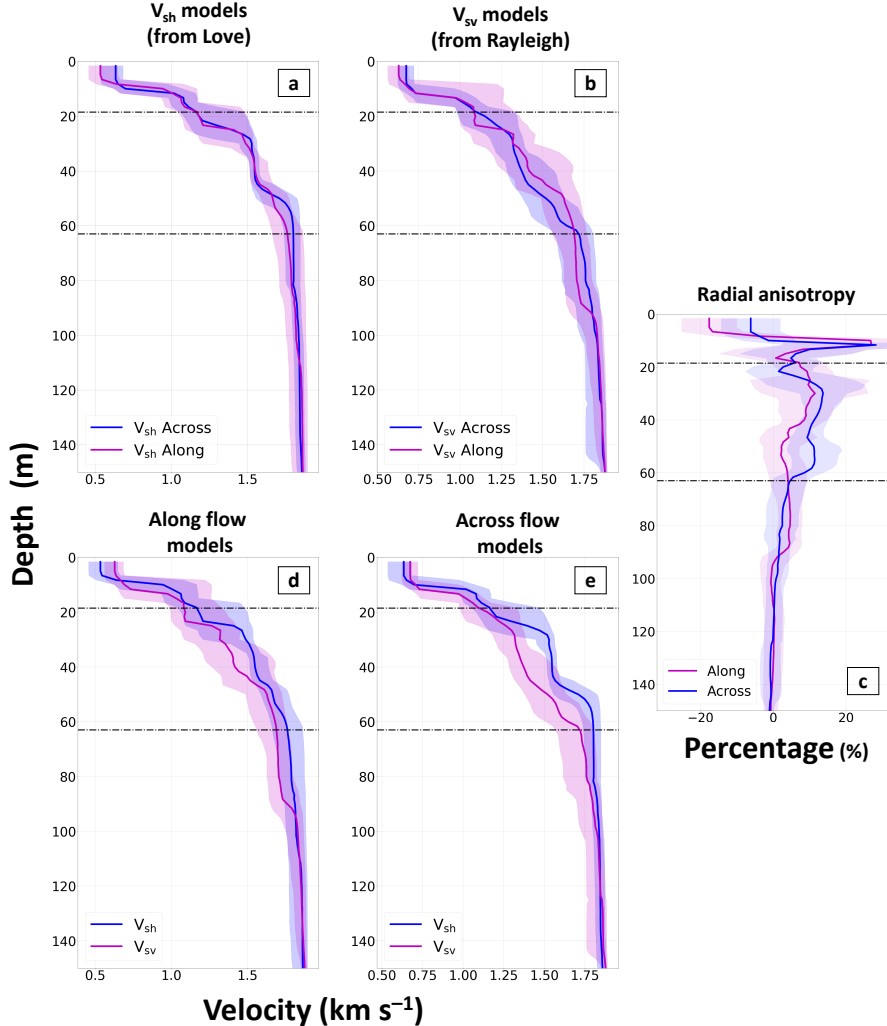

**Figure 7.** (a) $V_{sh}$ and (b) $V_{sv}$ models for the across (blue) and along (magenta) station pairs. (c) The percentage of seismic radial anisotropy $\left(\frac{vsh}{vsv} - 1 \times 100\right)$ for along flow stations (magenta) and across flow (blue). $V_{sh}$ (blue) and $V_{sv}$ (magenta) for (d) along flow (e) across flow. All figures include black dashed lines at the critical density depth (18.5 m for 530 kg m$^{-3}$, and pore close off (63 m for 830 kg m$^{-3}$) taken from the 2018 S6 ice core. The shaded regions are plus and minus one standard deviation from the most likely model output from the MCMC inversion.

## 5 Discussion

### 5.1 Seismic Anisotropy

Seismic anisotropy in firn is caused by a combination of different processes: (1) effective anisotropy, caused by thin layers
in the firn with different seismic velocities, smaller than the seismic wavelength, causing effective anisotropy. (2) Intrinsic



anisotropy, caused by the preferred orientation of the hexagonal ice crystals, the Crystal Orientation Fabric (COF) and (3) structural anisotropy (e.g. crevasses, micro-cracks). In the following, we discuss our results with respect to each of these potential causes.

### 5.1.1 Effective Anisotropy

The effective bulk anisotropy is an anisotropy caused by layers with varying density, i.e. layers with varying seismic velocities, significantly thinner than the seismic wavelength (Backus, 1962). At the NEGIS, firn density varies at a minimum, on the meter scale (Vallelonga et al., 2014), and our wavelength has a minimum length of approximately 40 m. Thus, the seismic wave experiences an effective, averaged medium velocity. From ice cores obtained at EastGRIP (Fig. 6), layering of different densities a few centimetres thick caused by buried dunes or sastrugis occur frequently in the core and are the main cause of the

layered structure in firn. Additionally, wind crust layers (1 mm thick) are seen, along with melt layers. The layers produce a vertical transversely isotropic (VTI) medium, i.e. anisotropic with a vertical axis of rotation symmetry.

In a VTI medium, SH-wave velocities and SV-wave velocities are not equal, with their velocity dependent on the angle of incidence. Rayleigh wave velocity, depending on the $V_{sv}$, and Love wave velocity, depending on $V_{sh}$, reflect this effective anisotropy. Consistently, we see $V_{sh}$ greater than $V_{sv}$ due to this effect.

From our analysis, the difference in $V_{sh}$ and $V_{sv}$ is a maximum of 15% at a depth of 30 m and 12% at 30 m for across and along flow respectively. This effective anisotropy in firn due to thin layers was observed previously in Antarctica (Diez et al., 2016; Schlegel et al., 2019). We compare this to anisotropy measurements obtained from active seismic experiments from the Antarctic plateau (Schlegel et al., 2019), where they recover a maximum anisotropy of 16%, and from passive data on an Antarctic ice shelf (Diez et al., 2016) where 15% higher velocities between 12 m and 65 m were seen for $V_{sh}$ compare to $V_{sv}$.

The difference in the magnitude of the radial anisotropy between the along and across flow can possibly be attributed to the predominant wind direction at EastGRIP. The average predominant wind direction in 2022 for EastGRIP was 273° (EastGRIP, 2023), approximately 17° from the across flow direction, versus 70° for the along flow direction. The extent of high-density layers in the firn originating from buried dunes are typically enlarged along the predominate wind direction as the dunes at the surface are enlarged (Birnbaum et al., 2010). As such, you observe stronger density variations in the predominant wind

direction. This leads to a greater difference in $V_{sh}$ and $V_{sv}$, and hence a larger radial anisotropy is seen in the across flow direction.

### 5.1.2 Intrinsic Anisotropy

In addition to effective anisotropy caused by small layers in the firn, we observe a relationship between changes in the magnitude of the radial anisotropy and variations in the intrinsic structure of the firn. Intrinsic anisotropy is caused by a preferred

orientation of the anisotropic, hexagonal ice crystals and has been observed in seismic data previously (Picotti et al., 2015). The intrinsic anisotropy develops due to the stresses and accumulated strain within the ice, which are comparatively small within the firn. In general, the crystal fabric in the firn is expected to be rather isotropic, with slight preference for vertical single maximum in the absence of other strain than compaction only. At the NEGIS, however, the few available measurements





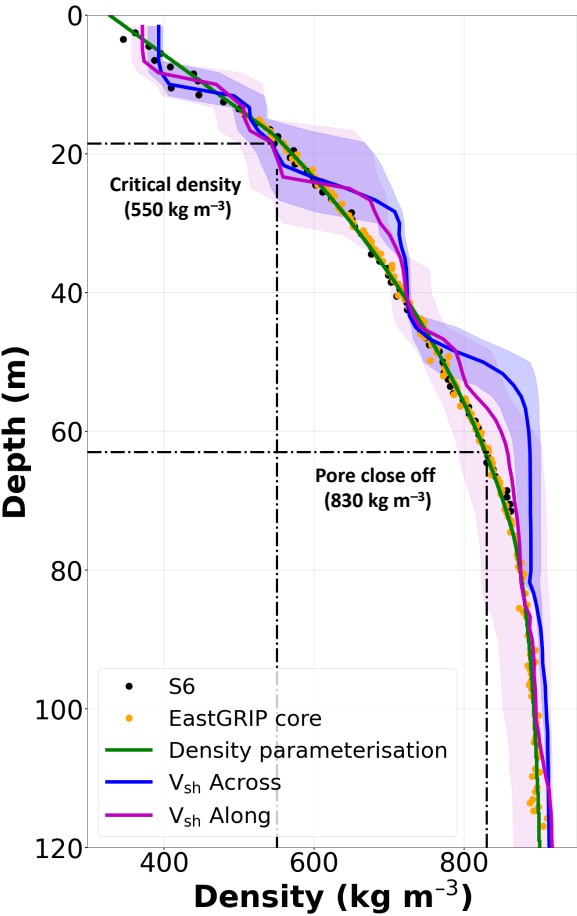

**Figure 8.** Velocity models converted to density using Diez et al. (2016) empirical scaling law, and compared to density cores recovered from the NEGIS (S6 and EastGRIP core). Dashed lines at the critical density depth (18.5 m for 530 kg m$^{-3}$, and pore close off (63 m for 830 kg m$^{-3}$) taken from the density parameterisation of EastGRIP.

of fabric at a depth of around 100 m indicate that, already at that depth, a weak single-maximum anisotropy has developed.

Below 100 m, this begins to transit towards a girdle fabric, increasing with depth (Gerber et al., 2023).

Using the empirical relationship from Diez et al. (2016) we convert our $V_{sh}$ models to density (Fig. 8) and compare the results to the depths at which we see the transition from the critical density (550 kg m$^{-3}$) at 18.5 m and pore close off (830 kg m$^{-3}$) at 63 m (Fig. 7), taken from the density parameterisation of EastGRIP.

In the first stage of densification (0–20 m depth), the dominant mechanism is grain settling and packing, i.e. the displacement

of grains. This is usually the most rapid stage of densification and ends when the density reaches the critical density at 550 kg m$^3$, around a porosity of 40% (Cuffey and Paterson, 2010). At this point, the grains have compacted as much as possible by displacement, and other mechanisms become active to cause any further densification. Here, we see no distinct pattern in the




anisotropy with $V_{sv}$ and $V_{sh}$ switching between being larger or smaller. As mentioned, we have no sensitivity to the upper 20 m of our models, hence, we can make no conclusions about the radial anisotropy in this part.

At a depth of 20 m, the critical density (550 kg m$^{-3}$) is reached, and there is a decrease in densification rate as the dominant process now becomes recrystallisation. As the shape and size of the crystals change to reduce the stress exceeded on the grain boundaries, a combination of molecular diffusion and movement along inner planes sees pore space reduce further and density increase to a value of around 830 kg m$^3$ (Herron and Langway, 1980; Faria et al., 2014a, b).

We see this reflected in the seismic radial anisotropy (Fig. 7 c), where the maximum anisotropy (12%–15%) is observed after the critical density is reached, steadily decreasing until a density of 830 kg m$^{-3}$ (pore close off). Thus, the recrystallisation of the firn from the increased compaction has an effect on the radial anisotropy.

### 5.1.3 Structural Anisotropy

At EastGRIP crevasses were not noted during previous geophysical surveys, on satellite images nor during any other field activity in the area surrounding EastGRIP, and field studies of the area report that EastGRIP is a field site without crevasses (Ice and Climate Group, 2023). Though we cannot exclude micro cracks from having an impact on the seismic anisotropy, there is no evidence to suggest these structures are present.

### 5.2 Firn thickness at NEGIS

We choose to use our $V_{sh}$ model to estimate the firn thickness since the equation from Diez et al. (2016) is based on SH-waves, using a velocity of 1800 m s$^{-1}$ (approximately that of ice, (Peters et al., 2008)) and a density of ice at 921 kg m$^{-3}$, as assumed in the EastGRIP density parameterisation.

The along flow velocity model shows a firn–ice transition between 60–70 m. This estimation is in agreement with other studies carried out in the ice stream, such as Fichtner et al. (2023), at between 65 and 71 m, Vallelonga et al. (2014) at approximately 65 m, and Riverman et al. (2019), at 68 m–70 m. Furthermore, it closer matches the firn–ice transition observed in the density parameterisation of the EastGRIP borehole at 63 m.

The across flow density model (Fig. 8) has a firn–ice transition at a depth of 50 m, thus showing increased densification than the along flow profile, for which the firn–ice transition is 10 m deeper. We attribute this measurement to the increased densification from the predominant wind direction at EastGRIP. Additionally, we expect to see firn thickness decrease towards the shear margin, due to increased accumulation (Riverman et al., 2019) and strain softening (Oraschewski and Grinsted, 2022), where firn thickness decreases to a minimum of 40 m.

These interpretations are based on the most likely models obtained from our MCMC inversion, however at the depth of the firn–ice transition, we observe that the uncertainty in our results could lead to firn being a similar thickness regardless of direction. However, the uncertainty in this region still indicates that the across flow model always has a higher density at depth than the along flow model, hence indicating increased densification.

Riverman et al. (2019) observed homogeneous firn above 35 m, regardless of the proximity to the shear margin. This seems also to be reflected in our results, where above 40 m, we see marginal variation between the along and across flow $V_{sh}$ models.





The information into the 2-D coverage of firn gained from 29 days of ambient seismic noise correlations provides additional insight into the glaciological context of the NEGIS and further information about the recrystalisation process' of firn. Our results suggest an increased densification of firn across flow due to the predominant wind direction and the increasing proximity to the shear margin. Additionally, the processes that dominate after the critical density (550 kg m$^{-3}$) is reached increase the

magnitude of the radial anisotropy of the firn structure. A combination of effective and intrinsic anisotropic effects leads to variations in radial anisotropy in firn across and along flow. This has implications for firn structure and rigidity, where the process of recrystalisation causes horizontally polarised shear waves to travel faster than vertically polarised shear waves. Furthermore, it shows the anisotropy effects in the firn are not only caused by effective bulk anisotropy, but also by intrinsic anisotropy. At present, seismic studies in the NEGIS use anisotropy values of 5% (Fichtner et al., 2023), but we show that at

EastGRIP camp, it is much higher. Furthermore, seismic studies in the NEGIS used to obtain estimations of firn thickness (e.g. Riverman et al., 2019), require measurements of the anisotropy in the firn to improve the accuracy of modelling.

### 5.3    Stability of correlations with time

We assess the signal-to-noise ratio of our cross-correlations for all station pairs and assess the improvement in SNR with the amount of days stacked (Fig. 9). We stack our data for a maximum of 23 days, after which only 8 of the 23 stations were

recording. We are able to show that for 23 stations, by nine days of recording, the improvement in SNR from additional stacked days is minimal, suggesting our analysis can be replicated by a minimum of nine days of noise recording. The high-frequency content, dense station spacing and high scattering rate of snow and ice allow stable imaging results with shorter recording times.



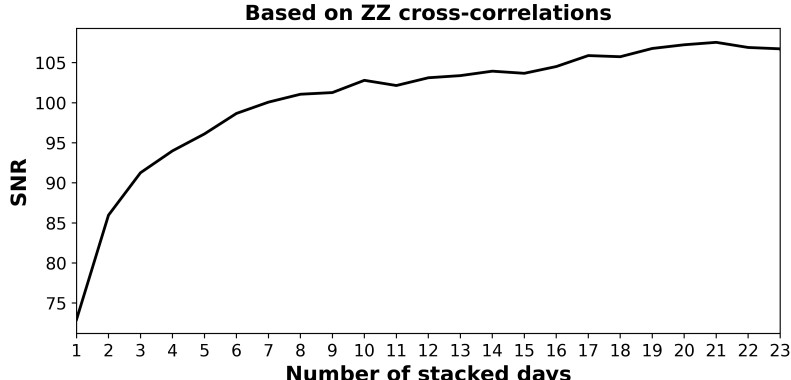

**Figure 9.** Mean signal-to-noise ratio of cross-correlation functions between all station pairs versus the number of stacked days. The results reach 95% of the maximum SNR after 8 days.

## 6 Conclusions

We demonstrate that simple, short-duration, passive seismic deployment and environmental noise-based analysis can be used to determine variations in the 2-D structure of firn, thus allowing improved interpretations of changes in surface mass balance.

We observed variations in the magnitude of the radial anisotropy of the firn and the depth of the firn-ice transition along and across flow of the NEGIS. These results provide information about the 2-D variability of the ice stream related to the predominant wind direction and the proximity to the shear margin.

Additionally, we find a consistent relationship both along and across flow between the magnitude of the radial anisotropy and the firn densification process. We observe that the magnitude of the radial anisotropy is affected by the intrinsic processes of firn compaction where we see a distinct change in the magnitude of the radial anisotropy when the critical density (550 kg m$^{-3}$) and pore close off (830 kg m$^{-3}$) are reached.

These results obtained from cross-correlation of ambient seismic noise data present a promising methodology for obtaining 335 2-D models of firn coverage. The relative ease of deployment of seismic nodes and the accessibility to high numbers presents an exciting opportunity for glaciology to obtain insight into firn anisotropy and structure that has previously been limited to 1-D profiles from cores.

**Appendix A**



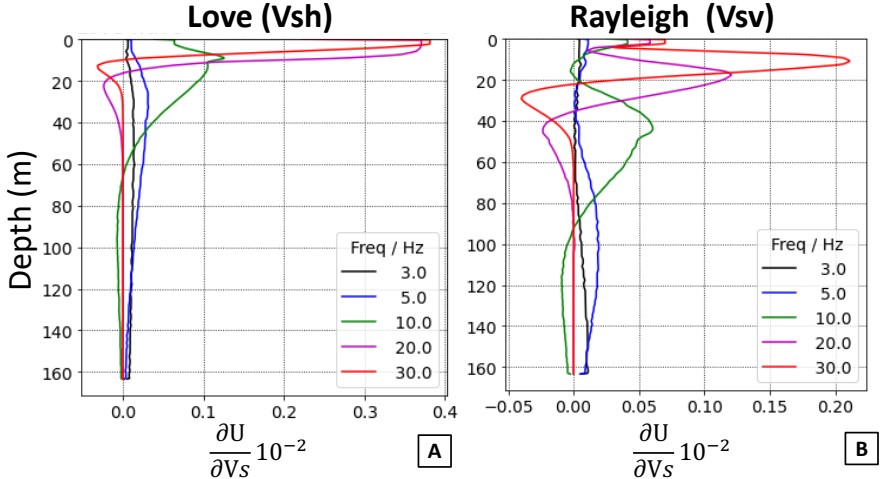

**Figure A1.** Depth sensitivity kernels at different frequencies for (A) Love and (B) Rayleigh for the fundamental mode. Note the differing axis between Love and Rayleigh.



**Table A1.** Frequency range and number of modes used for each model as the input for the MCMC inversions.

| Model | Frequency range (Hz) | Number of modes |
|---|---|---|
| Love: Along | 3.4-40.9 | 3 |
| Love: Across | 2.8-45.4 | 3 |
| Rayleigh: Along | 2.2–31 | 2 |
| Rayleigh: Across | 3-27 | 2 |

*Data availability.* Density data used for the 2023 parameterisation of the NEGIS can be found at PANGEA. Raw recorded seismic data for
the 23 stations can be found at PANGEA.

*Author contributions.* EP, DZ and OE conceptualized the study. CH and AF conducted the field work in 2022. EP, DZ and JR defined the methodology and processed the data. SOR and JF provided density data and the density parameterisation. EP, OE, JF and DZ interpreted the results. The manuscript was jointly written by EP, OE and DZ with inputs from all authors.

*Competing interests.* At least one of the (co-)authors is a member of the editorial board of The Cryosphere

*Acknowledgements.* The research carried out was funded by the University of Strasbourg Institute for Advanced Studies (USIAS) Fellowship to OE, which supported EP and equipment. Data were acquired at the EastGRIP camp in 2022. EastGRIP is directed and organised by the Centre for Ice and Climate at the Niels Bohr Institute, University of Copenhagen. It is supported by funding agencies and institutions in Denmark (A. P. Møller Foundation, University of Copenhagen), the United States (US National Science Foundation, Office of Polar Programs), Germany (Alfred Wegener Institute, Helmholtz Centre for Polar and Marine Research), Japan (National Institute of Polar Research
and Arctic Challenge for Sustainability), Norway (University of Bergen and Trond Mohn Foundation), Switzerland (Swiss National Science Foundation), France (French Polar Institute Paul-Emile Victor, Institute for Geosciences and Environmental research), Canada (University of Manitoba) and China (Chinese Academy of Sciences and Beijing Normal University). We thank the EastGRIP team for the great support in the field, in particular fearless driller Søren Børsting, Sverrir Hilmarson and Dorthe-Dahl Jensen. We also thank Maxime Bes de Berc who prepared the nodes, as well as Steven Franke, Nicholas Stoll and Fernando Valero Delgado for support, especially during the pilot study in
355 2019.



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
