# Peer review of "Firn Seismic Anisotropy in the North East Greenland Ice Stream from Ambient Noise Surface Waves"

_EGUsphere, 2023_

## Referee Comment (RC2)

**REVIEW**

This is an interesting paper where the authors show the presence of radial anisotropy in the firn from the analysis of ambient noise measurements. They obtained Rayleigh- and Love-wave dispersion curves from the computed cross correlations, picked the fundamental and higher modes and conducted 1D MCMC inversions. The obtained S-wave velocity curves show that Vsh is always higher than Vsv, both in the along flow and across flow directions. The authors conclude that radial anisotropy is present, and propose some explanation for that. The results are similar to what we found on the Whillans Ice Stream (WAIS), although we used active seismic data and other analytical methods (Picotti et al., 2017). Moreover, in our case we observed that the medium can be considered VTI, with the major differences in SH- and SV-wave velocities at the firn bottom, which we attempted to reproduce using Backus theory.

Overall, I find this work original, but the authors should be more critic about the approximations adopted and the uncertainties associated to the final results.

After reading the manuscript, I have the following major concerns:

1- Lines 125-129 : This is an interesting hypothesis, but not enough supported. Could you please add some references?

2- Lines 160-161 : "To avoid anomalous measurements, we further remove the frequencies for which the dispersion measurements look anomalous. The remaining values selected for the inversion are presented as orange curves". Could you please better explain the removal criteria?

3- The orange curves in Figures 4 and 5 represent the smoothed values used as the input for MCMC inversion. I noticed that in some cases these curves are quite different than the original picking of the maximum group velocities. Why? How much does this difference affect the final results? I have the impression that this mismatch is important.

4- Could you please explain why you used group velocities instead phase velocities (Figures 4 and 5)?

5- To my opinion data parametrization for MCTC could be better explained. Table 1 shows 6 layers and S-wave velocities ranging between 0.1 and 1.9 km/s for all layers. However, the authors do not justify these choices. In particular, why so wide ranges for the Vs at all depths? Have the authors considered to better constrain the S-wave velocities versus depth? For example, by using the density profile and the empirical relationship from Diez et al. (2016). Then, it is unclear whether the densities (Figure 6) were actually used in the inversion procedure. If density effects were ignored, how much this approximation affects the final uncertainties?

6- P waves were never mentioned in the article, which suggests that the contribution of P waves was likely ignored in the inversion. Again, how much this approximation affects the final uncertainties? Although Rayleigh waves weakly depend on P waves, I think that the authors should relate uncertainties of the inverted Vsv wave velocities also to the P to S wave velocity ratio. However, P waves can be easily modeled by using, for example, the density profile and the empirical relationship from Kohnen (1972), and can be included in the inversion.

7- The computed differences between Vsv and Vsh are small if compared to uncertainties. In the along flow direction, this difference is below the experimental error, with the error bars overlapping, i.e., the two curves are almost identical. Have the above approximations (i.e., ignoring density and P-wave effects) already considered in the estimation of uncertainties? The estimated weak anisotropy is already at the limit of detection threshold. My main concern is that even larger uncertainties may lead to indistinguishable Vsv and Vsh curves, and to a negligible anisotropy.

8- The error bars are larger between 20 and 60 m, and thinner above and below this depth interval. Moreover, errors are similar at surface and at the firn bottom. Since the deeper parts of the firn column are excited by the lowest frequencies, which have the largest uncertainties (the gray bands in Figures 4 and 5 are very wide below 5 Hz), one expects errors increasing with depth. Could the authors comment on that?

9-    Following points 7 and 8, the sensitivities shown in Figure A1 looks quite different for Rayleigh and Love waves. Thus, the inversion reliability is different for the two wave types and changes vs. depth. Could you comment on this? The authors should be more specific about the depth interval in which the estimation of the anisotropy is actually reliable.

10-   Lines 207-208 : "The terms of that covariance matrix are adjusted along the inversion to stabilize the acceptance ratio around 25%". Could you please expand this sentence please?

Minor comments

1 – Line 138: Signal-to-Noise ratio (SNR).

2 - Line 205 : (Herrmann, 2013) is redundant.

---

## Author Response (AR1)

**Response to Anonymous Reviewer:**

*My co-authors and I thank the reviewers and editor for careful analysis of our manuscript. In this following, we address point-by-point the concerns and outline the corrections we will make to our paper. We hope our replies and modifications are clear and satisfactory and we will get approval to go ahead with the revision.*

*Regards,*
*Emma Pearce on behalf of all co-authors*

Using ambient noise recording, the authors show seismic radial anisotropy of the firn layer in the North East Greenland Ice Stream. They pick the dispersion curves of Rayleigh and Love waves from the computed cross-correlations of ambient noise data and conduct 1D inversions. The difference between inverted Vsv and Vsh indicates the radial anisotropy of the target area. The results are similar to what we found in western Antarctica, although we used different inversion methods and focused on different areas. After reading the manuscript, I have the following concerns:

1. *Q. The source of ambient noise. As you mentioned, the EastGRIP camp may provide the primary source for ambient noise recording. From Fig. 1c, it seems the incident noise are more parallel to Line 1 other than Line 5. Did you observe the difference between the computed cross correlations for Lines 1 and 5?*

   A. The camp is not more parallel to line 1, only the runway has a direction and orientation which is orientated approximately 45 degrees between the two lines. Planes did not land when the ambient noise data were being recorded, we therefore assume that the camps orientation has no impact on the cross correlations. We did observe a difference between the computed cross correlations for line 1 and 5, but this is attributed to the variations in the ice stream. This is now clarified in the caption of figure 1.

2. *Following the first question, the crosscorrelations shown in Fig3 have strong energies at zero lag. You mentioned the possibility of wind. I would like to know whether wind could cause such strong energy and whether this could affect the calculated Rayleigh or Love waves.*

   A. The Zero lag cross correlations are attributed to wind since the previous season in 2019, seismometers were deployed without the use of bamboo and this feature is not seen. In the 2022 season, bamboos were placed less than 3 m from the same seismometers for relocation purposes and this feature appeared. Nothing else in the deployment was changed. The energy at zero lag is not used in the calculations of the cross correlations, a taper is applied to the data to remove the impact of this feature. Hence, no effect on the calculated love or Rayleigh waves is observed. In addition, this zero-lag feature has a frequency different from the one use to analyse the surfaces waves. This is now addressed in line 120.

3. *In the inversion, you use group velocity dispersion curves. How about the phase velocity? The picking shown in Fig 5 is misleading as nondispersive body S waves have been picked at high freqs.*

   A. We used group velocity to avoid dealing with 2 Pi jumps in the phase velocity estimations that are difficult to correct on noise cross correlations. The nondispersive S waves are indeed present in Fig. 5 and we agree with the reviewer that our figure is misleading. We have adjusted the figure and the caption to better explain the figure and the selected data to ensure there is no confusion.

4. *The radial anisotropy below 60m shown in Fig 7 is reaching zero. Is this caused by the reduced sensitivity of surface waves?*

A. In general, the sensitivity below 60 to 70m starts to decrease fast. We present the sensitivity diagram for Love and Rayleigh in appendix figure 1A and they show that for our available frequencies, we have sensitivity from our data to depths of 80-100 m based on those diagrams. Even with the reduced sensitivity at greater depths, the radial anisotropy begins to decrease when sensitivity is still high, and is lower after crossing the firn/ice transition.

5. *Fig 9 could be moved to the data processing section.*

   A. we thank the reviewer for their suggestion, but opt to keep figure 9 in the same location since it is independent of the noise correlation data processing.

6. *A typo of 'fig. AA1' in line 225.*

   A. We have altered this typo in the reviewed document.

7. *Line 210, 'an averaged 2D velocity' I'm confused since I only saw 1D profiles.*

   A. You are correct, the text has been changed to 1-D.

**Response to Stefano Picotti:**

This is an interesting paper where the authors show the presence of radial anisotropy in the firn from the analysis of ambient noise measurements. They obtained Rayleigh- and Love-wave dispersion curves from the computed cross correlations, picked the fundamental and higher modes and conducted 1D MCMC inversions. The obtained S-wave velocity curves show that Vsh is always higher than Vsv, both in the along flow and across flow directions. The authors conclude that radial anisotropy is present, and propose some explanation for that. The results are similar to what we found on the Whillans Ice Stream (WAIS), although we used active seismic data and other analytical methods (Picotti et al., 2017). Moreover, in our case we observed that the medium can be considered VTI, with the major differences in SH- and SV-wave velocities at the firn bottom, which we attempted to reproduce using Backus theory. Overall, I find this work original, but the authors should be more critic about the approximations adopted and the uncertainties associated to the final results.

My co-authors and I thank Stefano Picotti for your review and careful analysis of our manuscript. In the following, we address point-by-point your concerns and outline the corrections we will make to our paper. We hope our replies and modifications are clear and satisfactory.

Regards,
Emma Pearce on behalf of all co-authors

After reading the manuscript, I have the following major concerns:
1- Lines 125-129 : This is an interesting hypothesis, but not enough supported. Could you please add some references?

- A. We added references in the updated manuscript.

2- Lines 160-161 : "To avoid anomalous measurements, we further remove the frequencies for which the dispersion measurements look anomalous. The remaining values selected for the inversion are presented as orange curves". Could you please better explain the removal criteria?

A. This has been changed, and now reads "To avoid non representative dispersion measurements, we do not include modes higher than mode 3, since our dispersion curves at this point are not distinctive enough to establish which mode they represent. The criteria used to select the modes used were based on the gradient of the mode always decreasing."

3- The orange curves in Figures 4 and 5 represent the smoothed values used as the input for MCMC inversion. I noticed that in some cases these curves are quite different than the original picking of the maximum group velocities. Why? How much does this difference affect the final results? I have the impression that this mismatch is important.

A. We address this point in the resubmission by presenting better the dispersion curves and the data used, where it is now clear how the orange curves closely represent the raw dispersion measurements. The choice to use the orange smoothed dispersion measurements is due to the slight variability in the dispersion curves caused by stacking the measurements from each inter station pair. Furthermore, errors are included in the MCMC inversion which are representative of plus/minus one standard deviation between the maximum amplitude measurement from the dispersion curve.

4- Could you please explain why you used group velocities instead phase velocities (Figures 4 and 5)?

A. A. We used group velocity to avoid dealing with 2 Pi jumps in the phase velocity estimations that are sometimes difficult to correct on noise cross correlations.

*5- To my opinion data parametrization for MCMC could be better explained. Table 1 shows 6 layers and S-wave velocities ranging between 0.1 and 1.9 km/s for all layers. However, the authors do not justify these choices. In particular, why so wide ranges for the Vs at all depths? Have the authors considered to better constrain the S-wave velocities versus depth? For example, by using the density profile and the empirical relationship from Diez et al. (2016). Then, it is unclear whether the densities (Figure 6) were actually used in the inversion procedure. If density effects were ignored, how much this approximation affects the final uncertainties?*

A. The choice of velocity range for the inversion is based on the range of possible velocities that exist for snow and ice. The wide range is kept to avoid prior constraints on the inversion. Since the transition of firn to ice is a relatively fast process with a large velocity gradient, it is important to allow all possible velocities for depths in order to not pre condition the inversion. This is now explained in the Table caption. Densities were not used during the inversion since it is not possible to obtain them independently prior to the inversion. If Vp were recorded from refraction seismic, then this could be used to pre condition the density measurements, but in this instance, that was not the case.

*6- P waves were never mentioned in the article, which suggests that the contribution of P waves was likely ignored in the inversion. Again, how much this approximation affects the final uncertainties? Although Rayleigh waves weakly depend on P waves, I think that the authors should relate uncertainties of the inverted Vsv wave velocities also to the P to S wave velocity ratio. However, P waves can be easily modeled by using, for example, the density profile and the empirical relationship from Kohnen (1972), and can be included in the inversion.*

A. In our inversions with actually invert for Vp/Vs ratios. The results show large errors bars for that parameter as the sensitivity of the Rayleigh waves to P waves is low as state by the reviewer. As a result, constraining P wave velocity with our dataset is difficult. Even if this could potentially be overcome by incorporating prior knowledge on Vp or by modelling P waves by assuming a density profile, we don't have such independent information available. Note that our Vsv models fit well to the Fichtner et al. (2023) Vsv model that was obtain by recording Rayleigh waves produced by an airplane landing with DAS on a fibber optic cable (blue curve on the figure below). The slight variations between Fitchner et al. (2023) model and ours could come from the different frequency content used (more High Frequency for DAS) and the fact that the orientation of the DAS cable compared to the ice flow is not the same as the nodes arrays that we use in the present study. However, despite those slight differences, our Vsv models agree well with Fitchner et al. (2023) model and all of those Vsv models are clearly slower than the Vsh models confirming the radial anisotropy discussed in our manuscript.

[Figure]

*7- The computed differences between Vsv and Vsh are small if compared to uncertainties. In the along flow direction, this difference is below the experimental error, with the error bars overlapping, i.e., the two curves are almost identical. Have the above approximations (i.e., ignoring density and P-wave effects) already considered in the estimation of uncertainties? The estimated weak anisotropy is already at the limit of detection threshold. My main concern is that even larger uncertainties may lead to indistinguishable Vsv and Vsh curves, and to a negligible anisotropy.*

A. As describe in the previous point, we do have and independent Vsv model from Fitchner et al. (2023) which fit well with our Vsv models. In addition, thanks to the very hight density of the DAS measurements the uncertainties of the Fitchner et al. (2023) are lower than in our models. We are therefore confident that our Vsv models are reasonably good. As all those Vsv models are slower than our Vsh models, although the current uncertainty is at the limit of detection, we are confident in our interpretation that the variation is due to radial anisotropy and not uncertainty. Note also that the level of firn of anisotropy that we report here is typical for ice sheet/stream as stated in our paper and confirmed by both reviewers.

*8- The error bars are larger between 20 and 60 m, and thinner above and below this depth interval. Moreover, errors are similar at surface and at the firn bottom. Since the deeper parts of the firn column are excited by the lowest frequencies, which have the largest uncertainties (the gray bands in Figures 4 and 5 are very wide below 5 Hz), one expects errors increasing with depth. Could the authors comment on that?*

A. The error bars are dependent on the number of measurements at each frequency and the consistency of those measurements. It is true that the error are relatively large for low frequencies, however, despite those large errors, most models converge toward the 1800 m/s S waves velocity because such velocities are needed to fit the sharp increase of surface waves velocity at low frequencies. As a results, the error bars at low frequencies are small. The middle part of the models have the largest uncertainty as this is the region where there are large velocity variations over a small depth, and hence of the 22,500 models that were ran as part of the MCMC inversion, this region had the greatest variability in the modelled velocity.

*9- Following points 7 and 8, the sensitivities shown in Figure A1 looks quite different for Rayleigh and Love waves. Thus, the inversion reliability is different for the two wave types and changes vs. depth. Could you comment on this? The authors should be more specific about the depth interval in which the estimation of the anisotropy is actually reliable.*

A. In a layered earth model, Rayleigh and Love wave phase velocities have different sensitivities in response to the change of S-wave velocity of the same layer. We now comment on this in the text adding an appropriate reference (Line 312).

10- Lines 207-208 : "The terms of that covariance matrix are adjusted along the inversion to stabilize the acceptance ratio around 25%". Could you please expand this sentence please?

A. References explaining this process are now added to this section.

Minor comments have been corrected
1 – Line 138: Signal-to-Noise ratio (SNR).
2 - Line 205 : (Herrmann, 2013) is redundant.

**Community comment**

This paper is interesting and well-written. The authors analyzed ambient noise seismic data and picked the Rayleigh and Love wave group velocities in order to obtain the Vsv and Vsh structures. More interestingly, they obtained the radial anisotropy and analyzed the possible causes of the specific feature. I thus recommend publishing this work after the authors address some issues further:

**Comment 1:** Please check Line 40; the reference by Pearce et al. (2023) seems to have nothing to do with refraction data.

A. The paper referenced uses refraction data to image Firn and hence is included for its relevance.

**Comment 2:** The core of this paper is the inversion of the picked Rayleigh and Love wave group velocities. As for the MCMC method, have you ever considered using the transdimensional MCMC method to solve the problem of priorly defining the layers? I mean, how to minimize the difference generated by models containing different layers?

A. We did not consider this in this instance and were happy with the MCMC method used. But will consider it for future research.

**Comment 3:** The authors did not clearly explain the Vp and density models used for inverison, which makes it hard for readers to validate the results obtained. Table 1 only contains layer thicknesses, numbers, and Vs velocities.

A. VP and density are not used for the inversion, but rather are obtained by converting the Vs output of the MCMC inversion. We now ensure it is clear that this is how we obtain the density model. We do not comment on Vp in our data.

**Comment 4:** How to convince readers that the retrieved difference between Vsv and Vsh is not caused by inversion uncertainty. I mean, if we perform the MCMC method multiple times, is the difference between Vsv and Vsh still the same or similar?

A. This is always something which might be possible, no matter if at 10, 1 or 0.1% confidence level. However, as we discussed the uncertainties we come to the conclusions as pointed out in the paper. The MCMC inversion runs for a total of 22,500 models, and uses the likelihood

from each of these models to find the most probable solution. Hence, we are content that the difference between Vsv and Vsh remains.

**Comment 5:** *Please explain why the second-order information in Figures 5b and 5d ill-fitted the picked ones.*

A. The modes are forward modelled using a best guess understanding of the subsurface velocity. Therefore they are not able to perfectly match the true observed modes. After we have modelled the data using the MCMC inversion, this matching of the modes is improved. We use only the best guess forward modelled modes to identify whether the modes we observe are the fundamental, first or second order.

**Comment 6:** *I suggest the authors perturb the velocity of Vsh and Vsv within 0~20 and 60~140, like 10%, to see whether the sensitivity exists or not (comparing the calculated group velocities).*

A . Thank you for your suggestion. We allow the MCMC inversion to explore a range of velocities between 400 and 1800 m/s, and therefore this has been accounted for in the modelling.

**Comment 7:** *I recommend the author calculate the Vp/Vs ratio to gain more insights (if possible).*

A. This is not possible since we do not obtain any form of constained Vp model from the MCMC inversion.  This is now better explained in the text that we are not sensitive to Vp.

---

## Author Response (AR2)

Dear Stefano,

Myself and my co-authors thank you for your detailed review of our paper, and taking the time to re-review it. We acknowledge your recommendations, and hope the following adaptations to our paper satisfy your requests.

Many thanks,
Emma Pearce, and co-authors.

**Review from Stefano:**

The authors have made a good effort to address the questions raised during the first review. However, there is still some improvements to be made before publication.

I encourage the authors to include a more detailed version of their response to point 7 (and 6) of my previous review, in the manuscript. This point is very important. Currently, it is not yet clearly specified in the manuscript that density and P-wave velocity contributions are ignored in the inversion procedure, making it hard for readers to validate the obtained results.
A. An additional two paragraph has been included from line198 and  236, discussing the limitations of our P-wave velocity and density measurements obtained from the MCMC inversion, and the impact of ignoring these contributions.

Moreover, there is still no discussion in the paper about the effects of this choice on the estimated uncertainties. I believe this discussion is necessary to demonstrate that the claimed, very weak anisotropy, is real and not merely apparent. In other words, the authors should be more convincing about the reliability of their results.
A. further paragraph from line 263 in the discussion is now included examining the effects of the choice of not including Vp on the estimated uncertainties. This paragraph also includes validation of our results in comparison to Fichtner 2023 velocity model from the same vicinity at EastGRIP. A further figure has been included in the supplementary material showing this result comparison and aiding in the validation of our results for the reader.

**Minor comments:**

1- Figure 4 and 5 – Because of the grey error bands, I suggest to use a different color to represent the dots indicating the forward-modeled dispersion curves.
    A. We have adjusted the colour of the dots from grey to blue to be more visible on the figures.

2- Line 210 – A reference is missing (question mark "?")
    A. This is a typo and should only be the single reference. The (?) is now removed.

---

## Author Response (AR3)

Dear Dr. Horgan and the Editorial Team,

Thank you for accepting our paper for publication, pending the correction of a few minor revisions. In the revised document now uploaded, I have addressed all the technical comments provided and included the appropriate references where requested. Additionally, I have consolidated the concluding remarks into a single cohesive paragraph.

I hope these changes meet your expectations and that the paper is now ready for publication. I greatly appreciate all the hard work, time, and patience you and your team have dedicated to this paper over the past year.
Thank you again for your support.

Best regards,
Emma Pearce